# Object-Centric Learning of Neural Policies for Zero-shot Transfer over Domains with Varying Quantities of Interest

## Abstract

Our goal is to learn policies that generalize across variations in quantities of interest in the domain (e.g., number of objects, motion dynamics, distance to the goal) in a zero-shot manner. Recent work on object-centric approaches for image and video processing has shown significant promise in building models that generalize well to unseen settings. In this work, we present *Object Centric Reinforcement Learning Agent (ORLA)*, an object-centric approach for model-free RL in perceptual domains. ORLA works in three phases: first, it learns to extract a variable number of object masks via an expert trained using encoder-decoder architecture, which in turn generates data for fine-tuning a YOLO-based model for extracting bounding boxes in unseen settings. Second, bounding boxes are used to construct a symbolic state consisting of object positions across a sequence of frames. Finally, a Graph Attention Network (GAT) based architecture is employed over the extracted object positions to learn a dense state embedding, which is then decoded to get the final policy that generalizes to unseen environments. Our experiments over a number of domains show that ORLA can learn significantly better policies that transfer across variations in different quantities of interest compared to existing baselines, which often fail to do any meaningful transfer.

## 1 Introduction

Deep reinforcement learning (RL) has achieved significant success in learning policies for unstructured environments, e.g., where a state is input as an image. Indeed, Deep RL has often surpassed humans on a number of such problems, such as Atari games (Silver et al., 2016). On the other hand, the task of learning policies that can generalize to unseen environments can be particularly challenging (Kirk et al., 2021). In this paper, we take a step towards strengthening Deep RL's zero-shot transfer capability, by studying the setting of transfer over *quantities of interest*.

Even if a (high-level) domain has an unstructured state, each of its environment variations may be characterized by various (latent) quantities of interest (QoIs), such as the number of objects of each type (e.g., the number of legs of a centipede, number of balls/bricks in the game, number of enemies to kill), the numerical attributes of each object (e.g., position or velocity of each object, the maximum force of every muscle), or other derived features that make the instance easy or hard (e.g., distance of the robot to the goal). Our goal is to study transfer across these QoIs. In particular, given a small set of learning environments governed by a set of QoIs, how do we effectively generalize to environments with unseen (generally higher) values of these QoIs? Figure 1 shows an example domain, where the transfer is over the number of balls in game playing.

A natural question arises: what might be an effective model for this problem? Our premise is that existing RL models which work directly on the perceptual input, oblivious of the object representations, may not generalize - a thesis confirmed by our experiments. We take recourse to recently developed object-centric models for images and videos (Burgess et al., 2019; Kossen et al., 2020; Locatello et al., 2020; Wu et al., 2022), which have shown a lot of promise in achieving better interpretability, robustness to noise, and ability to generalize to unseen settings. Motivated by these works, we use the object-centric representations for the

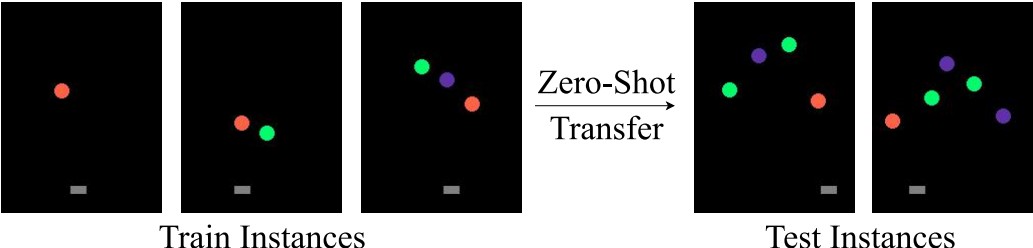

Train Instances                                    Test Instances

Figure 1: Given interactions in environments with 3 or fewer balls, with the goal to keep the balls from falling, our objective is to learn a policy which would do well in an environment with four or more balls.

task of model-free RL and propose an object-centric approach for learning neural policies, which can transfer well across QoIs in the underlying environment in a zero-shot manner.

Our model, referred to as Object-centric Reinforcement Learning Agent (ORLA), works in three phases. (1) We extract object masks that encode object positions in a sequence of frames. This is done using a novel pipeline where we first train an expert using an auto-encoder-based architecture where the object masks are regularized by a Gaussian distribution. The data from the expert is used to do supervised fine-tuning of a YOLO (Jocher et al., 2022) based model for learning object bounding boxes. This model has the capability to generalize to a variable number of objects. (2) The bounding boxes output by YOLO are used to retrieve object positions in each frame to build an *object-centric* state representation for the underlying MDP. (3) This state representation is passed through a Graph Attention Network (Veličković et al., 2018) (GAT) to learn a dense state embedding, which captures object interactions. Each object node in GAT maintains numerical attributes of that object (e.g., position, velocity) as explicit features, enabling better transfer over QoIs that depend on such attributes. The dense embedding is then passed through an action decoder network to output the policy in the current state. This policy naturally generalizes to unseen environments, due to the inductive nature of GATs.

We design two different categories of domains to test ORLA compared to CNN-based agents and other relevant baselines. In the first category of domains, we study game-playing with balls and pedal, where we transfer across the number of balls, and motion dynamics in two different settings: (i) vertical ball movement and (ii) diagonal ball movement. In the latter case, we also test with balls having different behavioral characteristics. In the second category of domains, we study navigation in environments with (i) unseen distance from the goal, and (ii) variable number of sub-goals in an adaptation of the traveling salesman problem. Our results indicate the superior performance of our approach compared to baselines, which often fail to do any meaningful transfer.

In summary, we make the following contributions. (1) We study transfer in RL task, (a) on domains that are observed as unstructured images, even if they have underlying objects and other structures, and (b) for transfer to domains that vary quantities of interest, such as the number of objects or other numerical attributes of the objects/domain. (2) We present an object-centric RL formulation where the image is converted into a symbolic object representation, and the RL agent is trained on top of the symbolic representation. (3) We construct new domains and experimental settings to study this task. (4) Through a series of experiments, we find that our object-centric RL is able to effectively achieve the required transfer to target domains, whereas other baselines are not able to do any meaningful transfer. We will release the code and environments for future research on acceptance.

## 2 Related Work

**Object-centric Representations:** MONet (Burgess et al., 2019) learns to decompose scenes into various objects using an unsupervised loss. COBRA (Watters et al., 2019) uses MONet to learn a transition model of a continuous control environment to be used in model predictive control. Though similar to our work in spirit, their architecture does not support interactions among objects and does not do model-free RL. STOVE (Kossen et al., 2020) uses variational autoencoders to explicitly predict the object positions for

learning the dynamics of the environment rather than the policy. Another series of works (Kipf et al., 2020; Locatello et al., 2020; Wu et al., 2022) learn object-centric representations using slot attention for learning the dynamics of the environment, but do not focus on learning the policy.

**Transfer Learning in RL:** Earlier works use domain randomization for generalization in RL, focusing on how to sample a set of training environemnts from an underlying distribution, hoping that the test instance will be close to the training environment (Kirk et al., 2021). Another line of work takes an object-centric approach; for example, NerveNet (Wang et al., 2018) studies the size and disability transfer tasks by formulating an agent as a graph and learning a Graph neural network-based policy. Shared Modular Policy (Huang et al., 2020) (SMP) tries to learn a shared policy for multiple agent morphologies. However, both NerveNet and SMP assume that objects, their properties, and whether they affect each other or not are explicitly provided. That essentially means that nodes, node features, and edges of the graph are provided by the environment. This significantly restricts their application in comparison to ours, where we extract these from raw image frames in a self-supervised manner. Zambaldi et al. (2019) also do experiments on the size-transfer task by viewing the output of a CNN as a set of entities and using attention to learn a policy; however, they do not predict the positions of the objects explicitly, and the entities do not directly tie to a specific object in the scene. COBRA (Watters et al., 2019) also studies transfer learning but takes a model-based approach, unlike a model-free one, which is the primary focus of our work. Indeed, we can first learn a object-centric world model (e.g. STOVE) and then learn a policy on top of it. In this work, we restricted ourselves to the model-free approach and studying this like of work is left for future work.

**Generalized Neural Policies in Planning:** Learning generalized policies that can solve any instance of a domain is a well-studied problem in stochastic relational planning (Srivastava et al., 2008; Hu & De Giacomo, 2011; Belle & Levesque, 2016). A series of recent works (Garg et al., 2019; 2020; Sharma et al., 2022; Ståhlberg et al., 2022; Sharma et al., 2023) learn generalized neural policies by converting an environment into a graph and then learning a policy to study the size-transfer task. In all these works the state is already represented in a symbolic language like Relational Dynamic influence Diagram Language (Sanner, 2010) (RDDL) and Probabilistic Planning Domain Definition Language Younes et al. (2005) (PPDDL). Further, given the planning setting in these works, they have complete access the environment's transition model. They leverage it to create a graph capturing dependencies among various state variables. In contrast, the state in our case is raw image(s), and we first learn to create a symbolic graph capturing dependencies among various objects in the scene (as we see in the next section).

## 3 The ORLA Agent

We are interested in learning generalizable neural policies that would work for zero-shot transfer over quantities of interest (QoIs) in a domain. In our setting, these QoIs may be latent, i.e., may not be directly accessible in the interactions with the environment, e.g., motion dynamics. Say, we are playing a game in an environment with perceptual input, with goal as moving the pedal to prevent the ball(s) from hitting the ground (Figure 1). We ask: if the agent is allowed to interact with environments with $k$ or fewer balls, and learns a policy over them, would it generalize to play the same game in an environment with greater than $k$ balls? Or equivalently, would the learned policy generalize when the motion dynamics in the game are different from those seen during training? We note that humans have a remarkable capability to achieve this kind of transfer, but it is not clear if modern-day AI agents can achieve this effectively.

A naïve approach would be to train a standard RL model, that takes in a sequence of frames and produces an action to be taken. These algorithms have been shown to do very well, often beating human players, such as in the Game of Atari. However, the task at hand is more challenging as the network has to learn to transfer across variations in QoIs. We first did a simple experiment to see whether a Dueling DQN-based agent (Wang et al., 2016) generalizes in such a setting. Interestingly, it fails completely on this task (see Experiments Section). So, we ask: what would be a better way of modelling such transfer tasks? What if we could directly learn the policy over a *symbolic* state representation, such as object positions and their interactions?

We deal with structured domains, consisting of objects of various types that can interact with each other. Each domain is characterized by some *quantities of interest* (QoIs), which are human interpretable. In this

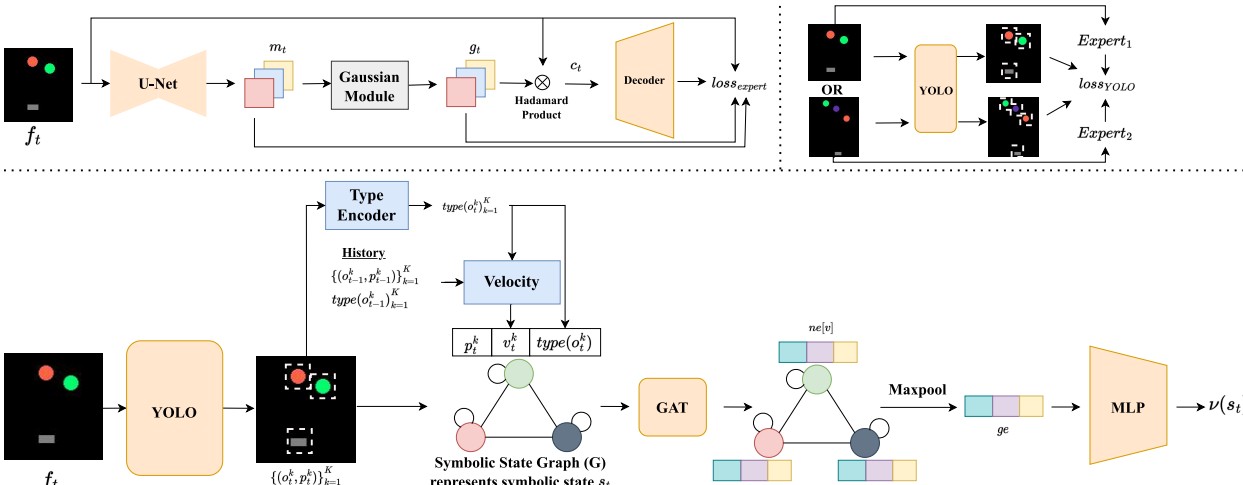

Figure 2: (Top Left) Training details of our unsupervised expert Object extractor. (Top Right) Training of YOLO-based object extractor using multiple Experts. (Bottom) A forward pass from ORLA that takes an image as input, converts it into a symbolic state graph, and then uses GAT-based architecture to get the policy.

setup, a domain can be thought of as a parameterized environment, where a specific assignment to the parameters instantiates a (ground) environment in which an agent interacts. Then, given a set of training environments of a domain, our objective is to learn a policy that is applicable to any environment of that domain.

Motivated by the earlier discussion, we take a three-pronged approach to attack this problem: (1) First, we extract object positions in a self-supervised manner without any object-level annotations. (2) The extracted objects are used to construct a symbolic state representation. (3) We learn a policy over the symbolic state in a manner that could generalize to varying values of QoIs. Next, we describe each of these in detail.

### 3.1 Self-Supervised Object Extraction

The aim of our Object Extractor (OE) is to extract the positions of each object in a given frame. Object extraction is a well-studied problem in computer vision, but most successful models require supervised data (Jocher et al., 2022; He et al., 2017), which is not directly available to us. Therefore, we pose the following question: can we use a supervised object extractor model, pre-trained on an existing dataset, and fine-tune it on our training instances in a self-supervised way? To this end, we propose a novel self-supervised training algorithm that generates a supervised dataset to fine-tune a pre-trained model for object extraction. Our algorithm first learns a set of *expert object extractors* (expert-OE), $\{Expert_1, ..., Expert_N\}$, where each $Expert_i$ learns to extract objects in the frames from the $i^{th}$ training environment in an unsupervised manner. We note that a single environment corresponds to a specific combination of values for QoIs, and we learn a different expert for each environment. Next, these experts are used to generate a single dataset of $\{(frame, objects)\}$, corresponding to all the training environments. Finally, a *supervised object extractor*, applicable on any environment of the domain, is fine-tuned using this dataset. We describe the details next.

#### 3.1.1 Environment Specific Object Extraction

The Expert Object Extractor (expert-OE) of an instance is an auto-encoder architecture that takes a frame of only a specific environment as input and locates objects in it. For this, it first disassembles a given image ($f_t$) into a set of constituting objects and then uses a decoder that composes these objects to form the original input image. There are three sub-modules: 1) An Attention U-Net (Oktay et al., 2018) that takes the image as input and returns a set of $K$ object masks (where $K$ is the maximum number of objects in the

environment), 2) A Gaussian module that enforces each object mask to look like a 2D Gaussian centered at the object's center, and 3) A Decoder that reconstructs the input image.

**Encoder:** Let there be an environment with $K$ objects in it. Given a frame $f_t$, the Attention U-Net gives $K$ object masks $\{m_t^i\}_{i=1}^K$ representing the $K$ objects in the instance (See Figure 2). We want the mean of each mask to represent the position of one of the objects. For this, we add a prior that each of the object masks should look like a 2D Gaussian with the mean at the centre of the object. The Gaussian mask ($g_t^i$) for each object mask $m_t^i$, is generated at the center of the object mask, i.e., with mean $(\sum_j j * \sum_k m_t^i(j,k), \sum_k k * \sum_j m_t^i(j,k))$ and standard deviation as $\Sigma^1$. Here, the mean of each $g_t^i$ represents the position of the $i^{th}$ object. This is similar to the Gaussian Module used in I-CSWM (Gupta et al., 2021). Next, to capture $i^{th}$ object's visual depiction (e.g. colour, texture, etc), we generate a content image as $c_t^i = f_t \otimes g_t^i$, where $\otimes$ is the Hadamard product.

**Decoder:** Our decoder regenerates ($\hat{f}_t$) the input image by combining the object contents $c_t^i$ at their respective positions (mean of the corresponding $g_t^i$) on a colored background[2]. For each training environment $i$, we collect a very small dataset $D_i$ of states seen by playing 25 episodes using a random policy (ref. Table 1 in the supplementary for exact details). We train each environment's expert-OE independently using the loss: $loss_{expert} = mse(f_t, \hat{f}_t) + \sum_{i=1}^K mse(m_t^i, g_t^i)$.

We also tried slot-attention (Locatello et al., 2020) as our expert-OE. However, in our experiments, it failed to bind one object per slot (see details in the supplement). Our expert-OE worked well for our domains, and developing better experts is left as future work.

### 3.1.2 Generalized Object Extraction

The goal of expert-OEs is to generate training data for finetuning a pretrained generalized object extractor. For each training environment $i$, we create a supervised dataset by labeling images in $D_i$ using the corresponding expert-OE i.e. $D_i^E = \{(f, Expert_i(f)) | \forall f \in D_i\}$. Finally, we finetune a YOLO (Jocher et al., 2022), pretrained on ImageNet, jointly over all environments using the dataset $D = \cup_i D_i^E$. We picked YOLO as it satisfies our requirement of transfer, i.e., it can extract a varying number of objects in the frame. For an input image in an environment, YOLO provides a set of bounding boxes around objects along with a confidence value. We pick the bounding boxes with the top $K$ confidence scores, where $K$ is the maximum possible number of objects in that environment. We define the position $p_t^k = [p_t^k[x], p_t^k[y]]$ of an object $o^k$ as the center of its bounding box.

We also need to assign a type to each object. For example, the balls and the pedal should be assigned a different type. We assume that the total number of types across all the environments in the domain is known apriori. Then, each object is assigned a type using k-means clustering over the extracted masks. Specifically, we assign $type(o^k) =$k-means$(YOLO(f_t)[o^k])$. During training, the clustering is done over the set of objects $\{Expert_i(f) \forall f \in D\}$, and assuming the number of clusters to be equal to the given number of object types in the domain.[3] During inference, we simply assign the type of object using the k-means.

## 3.2 Symbolic State Representation

Like conventional deep RL agents that stack the last few frames to incorporate the history, we incorporate the history in our symbolic state representation. However, since our OE gives an unordered set of objects, so it is not straightforward to provide the history; We first have to match each object $o^k$ at time $t$ to some object $o^l$ at time $t-1$ (called history-object). In our case, for an object $o^k$, we pick the object $o^l$ nearest to its position in the last time step, i.e., $l = \underset{k' \in L}{argmin}(dist(p_t^k, p_{t-1}^{k'}))$, $dist$ denotes the Euclidean distance, and $L$ denotes all objects at time $t-1$ that have type same as $o^k$. In our experiments, we found this simple method to work satisfactorily, and we leave the use of finer methods for future work.

---

[1]in expts, we assume a diagonal $\Sigma$ with $\sigma_x = \sigma_y = 2.5$.

[2]In our work, all our environments had fixed background color. However, we can easily relax this assumption by using the standard practice of subtracting the mean image of the dataset from $f_t$.

[3]An off-the-shelf method like Elbow method can be used to find no. of clusters if no. of object types is unknown.

Next, rather than simply stacking object positions at the last few time steps, we compute a velocity vector at time $t$. It computes difference in positions and direction of change for each object and is given as $v_t^k = [p_t^k[x] - p_{t-1}^l[x], p_t^k[y] - p_{t-1}^l[y], tan^{-1}(\frac{p_t^k[y] - p_{t-1}^l[y])}{p_t^k[x] - p_{t-1}^l[x]})]$, where $o^l$ is the history-object of the object $o^k$. Similarly, we can compute the acceleration of objects, but we found velocity to be sufficient for our experiments.

We organize the extracted objects in the form of a graph called *Symbolic State Graph (SSG)* representing a symbolic space, denoted by $G(V, E)$. Its vertex set $V$ contains a node for each object detected in the scene, and has a fully connected adjacency matrix, i.e., $E = \{(u, v) | \forall u, v \in V\}$. A node for an object $o^k$ has input features given by $[p_t^k \ || \ v_t^k \ || \ type(o^k)]$, where $||$ denotes the concatenation operator.

### 3.3 Learning a Generalized Policy

We will now discuss the details of our architecture to learn a policy in the symbolic state. ORLA can be trained using any standard deep RL algorithm (value-based/policy gradient/Actor critic) as discussed next.

**Backbone Network:** Our backbone network is primarily based on Graph Attention Network (Veličković et al., 2018). First, we compute a set of node embeddings ($ne$) by applying a GAT on $G$. For a vertex $v \in V$, $ne[v] = GAT(G)[v]$. Next, a global embedding representing a global view of the state is computed by feature-wise pooling over all node embeddings, i.e., $ge = maxpool_{v \in V}(ne[v])$, which is used to compute the value/policy as needed by the base RL algorithm. We trained ORLA using Dueling DQN (Wang et al., 2016) and PPO (Schulman et al., 2017) in our experiments.

**RL algorithm specific Network:** For dueling DQN, a feed-forward network (MLP in Figure 2) takes $ge$ as input and predicts the value of the state $\mathcal{V}(s) = mlp_{\mathcal{V}}(ge)$ and the Advantage function of an action $a \in A$ as $\mathcal{A}(s, a) = mlp_{\mathcal{A}}(ge)$. The q-value of an action $a \in A$ is computed as $q(s, a) = \mathcal{V}(s) + \mathcal{A}(s, a) - \frac{1}{|A|} \sum_{a' \in A} \mathcal{A}(s, a')$ where $A$ denotes the set of all actions. Similarly, for PPO, the MLP forks to predict the policy and the critic value.

**Training Details:** We adapt dueling DQN and PPO to train on multiple instances and keep a separate experience replay buffer for each training instance. Each training instance is picked in a round-robin fashion to generate an episode, where each frame $f_t$ is converted into its symbolic state $s_t$ and the $(s_t, a_t, r_t, s_{t+1})$ pairs of that episode are pushed into the instance's experience buffer. A batch is sampled from the instance's experience buffer to train the network. While training the network above, we freeze the weights of the YOLO. Experimenting with a jointly trained model is a direction for future work.

## 4 Experiments and Results

We compare four models in our experiments.

(1) **Random Policy (RND)**, which takes a random action with uniform probability.
(2) **Nature CNN (CNN)** represents a model with unstructured representation space. We use a nature CNN (Mnih et al., 2015) architecture with 3 CNN layers followed by dense layers and train it jointly on the set of training environments of a given domain.
(3) **Gold-GAT (G-GAT)** fetches the gold positions of objects from the environment and uses these, rather than the output of our generalized object extractor (YOLO), to train ORLA's network while keeping everything else exactly the same as in ORLA. While we use G-GAT as a baseline, a direct comparison to CNN or our approaches would be unfair, as the input state representations are different in these. We still include it to help us highlight the effectiveness of our object extractor as well as the policy networks, independent of each other.
(4) **ORLA**, for which we use the standard pretrained YOLO architecture (Jocher et al., 2022). We use a GAT with 2 layers. Our $mlp_{\mathcal{V}}$ has one hidden layer with 128 units and a single output unit. And, $mlp_{\mathcal{A}}$ has one hidden layer with 128 units and an output layer with units equal to the size of action space.

We refer the reader to supplementary for details on the architecture and training. We also tried the method proposed in Zambaldi et al. (2019). However, their code is publicly unavailable, and we did not get any

response from the authors. We tried replicating their architecture on our own, but it failed to train even on training environments for multiple hyperparameter settings.

We now discuss two case studies, each having two domains that capture various complexities of the underlying task to be performed. For all experiments, we report the zero-shot transfer reward on the test environment vs. #frames seen during training. We also show goal reachability on one domain. We will explain the results of dueling DQN in greater detail. For PPO, we observed similar trends and will highlight the deviations from dueling DQN (see supplement for a detailed analysis).

### 4.1 Case Study 1: Balls and Pedal

In this case study, we want to study our model's zero-shot transfer capabilities when QoIs capturing the size and motion dynamics vary in the test environments in comparison to what is seen in the training environments. For this, we create two domains having two types of objects (a set of balls and a pedal) that can interact with each other and have natural dynamics.

#### 4.1.1 Domain 1: Column-Balls (CB)

In the Column-Balls (CB) domain, the balls always move in a fixed column, and the pedal is controlled by the agent using three actions: left, right, and noop. The task is to reach a maximum score of +21 with a reward function given as +1 if the ball hits the pedal, -1 if it hits the ground, and 0 otherwise. We create five environments having 1, 2, 3, 4, and 5 balls, respectively. We train on environments 1-3 and test on 4-5. During training the column of the balls is fixed, but the starting y-point and the direction of movement (up/down) are sampled for each episode. However, during testing, we also sample the column for each ball along with its starting y-point and the direction (up/down) at the start of each episode.

**Size Transfer:** Figure 3(a) shows the results of the zero-shot transfer reward vs. #frames seen for test environments 4 and 5 when trained on environments 1-3. We can see that ORLA is able to generalize to unseen test environments with 4 and 5 balls with a very high reward. As expected, the GAT trained on gold positions (G-GAT) performs the best. ORLA performs only slightly worse than G-GAT, highlighting the efficacy of our object extractor. Interestingly, the CNN policy fails to generalize, performing only slightly better than the random policy (RND).

Looking at the rewards vs. #frames curves on the training environments (Figure 5 in supplementary), we observe that the CNN model achieves the best possible reward in all training environments however it fails to generalize to environments 4-5, highlighting the lack of generalization capabilities of the CNN model.

**Dynamics and size Transfer (Ball Direction):** Next, to study the dynamics transfer task, we create a domain called Diagonal-Balls (DB), where we allow the balls to move freely in any direction rather than just columns. Here, the QoI is the direction of the ball movement. We sample the start position and direction of balls at the start of each episode. We create two test environments of this domain with 4 and 5 balls each.

Figure 3(e) shows the results of this experiment where we compare ORLA, CNN, and RND policies. Interestingly, ORLA is able to do a zero-shot transfer with a very high margin in comparison to the CNN policy even when the underlying dynamics (movement direction and hence velocity direction) of balls are changed.

#### 4.1.2 Domain 2: Multitype Diagonal Ball

In the Multitype Diagonal Ball (MDB) domain, the pedal has to avoid certain balls while trying to hit some other balls, thus increasing the task's complexity as compared to CB. There is a paddle and two types of balls: red and green. The balls can move in any direction, which is sampled at the start of each episode. The task is to maximize reward in a fixed-length episode with a reward function given as +1 if the paddle hits the green ball, -1 if the green ball hits the floor, -1 if the red ball hits the paddle, +1 if the red ball hits the ground, and 0 otherwise. We create a total of 5 environments; the first one has 1 green ball, the second has 1 green and 1 red ball, the third one has 2 green and 1 red ball, the fourth one has 2 green and 2 red balls, and the fifth one has 3 green and 2 red balls. We train on environments 1-3 and test on 4-5.

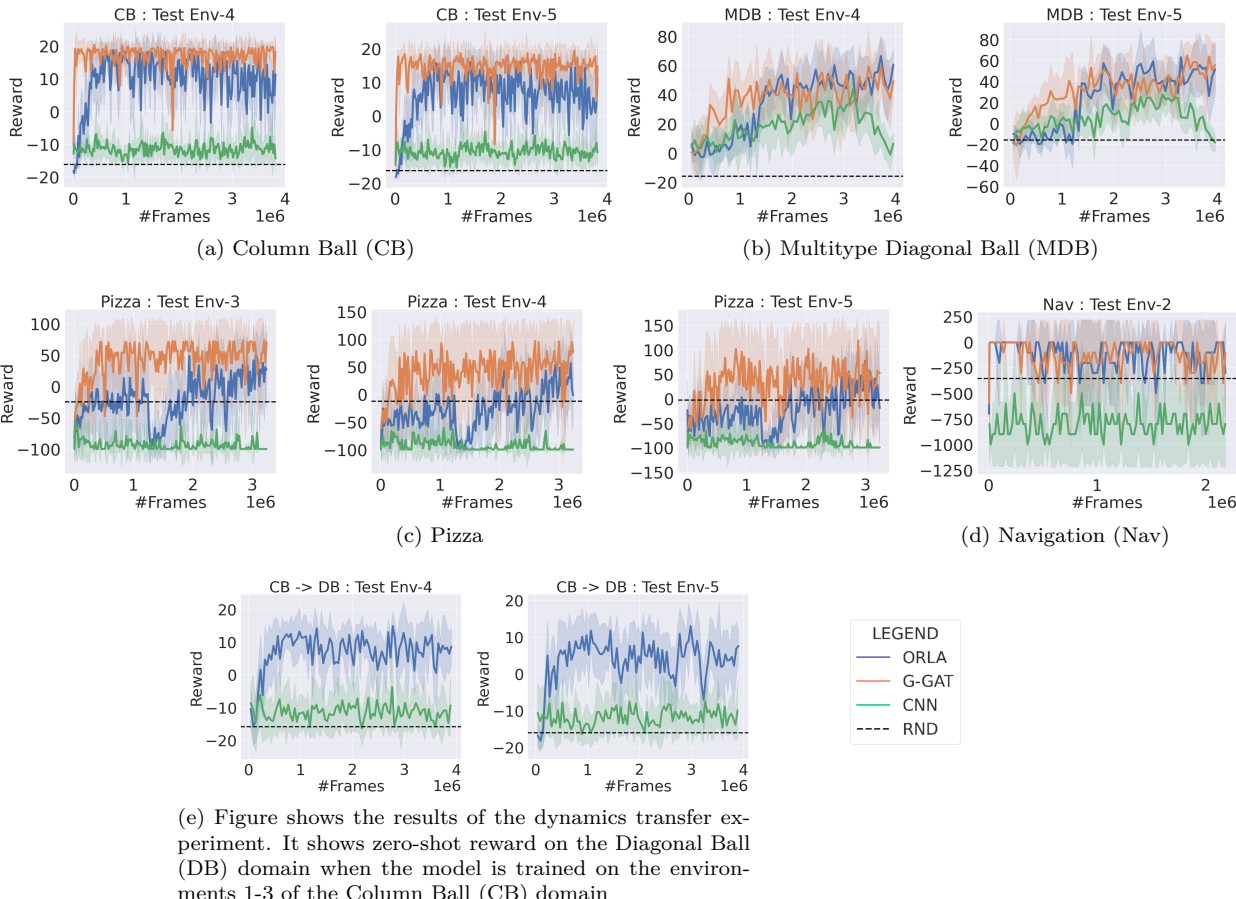

(a) Column Ball (CB)

(b) Multitype Diagonal Ball (MDB)

(c) Pizza

(d) Navigation (Nav)

(e) Figure shows the results of the dynamics transfer experiment. It shows zero-shot reward on the Diagonal Ball (DB) domain when the model is trained on the environments 1-3 of the Column Ball (CB) domain

Figure 3: (a-f) Figure shows the zero-shot transfer reward on the test environments vs. #frames seen on the training environments for all domains for dueling DQN. The dashed line represents the reward of a uniform random policy over 500 episodes to counter stochasticity in the policy. For all other methods, we take an average of 10 episodes. (See Figure 5 in the supplementary for graphs on all train and test environments of all domains.)

**Size Transfer:** Figure 3(b) shows the results of this experiment on the zero-shot size transfer task. Interestingly, ORLA performs equally well as compared to the G-GAT on both test environments. While CNN also performs better than RND, ORLA's performance always remains above CNN's.

We want to note that as domains CB and MDB have the same types of objects, we do not retrain our object extractor on MDB. Rather, we use the same extractor from CB, highlighting the modularity of our approach.

In summary, in this case study, we find that the object-centric approach of ORLA makes it much more amenable to transfer across both size of the domain (#objects) and numerical attributes of objects (direction of movement). In contrast, the vanilla CNN-based network tends to overfit the training environments and is unable to perform much meaningful transfer.

**Results on PPO:** Figures 6(a-b) in Supplement show results of PPO on CB and MDB. We notice that all methods show unstable training in comparison to their counterparts in dueling DQN. We attribute this to the inherent nature of PPO Schulman et al. (2017). This is evident from the performance of PPO-based baselines on the training instances. For example, consider the G-GAT baseline, which is the most stable among all baselines across all domains. Here, if we focus on the simpler of the domains, CB and MDB, the training curve for dueling DQN for G-GAT is much more stable as compared to when trained using PPO.

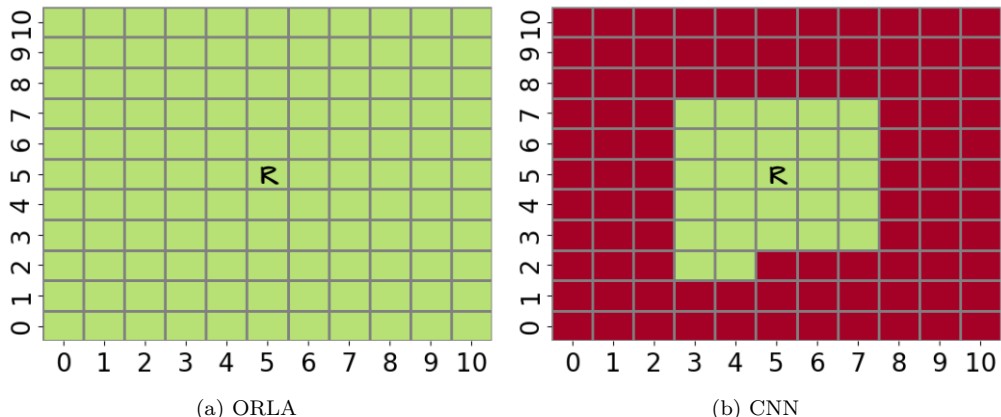

(a) ORLA                                      (b) CNN

Figure 4: Figure shows the goal coverage for best-trained models of ORLA and CNN on the Navigation domain when trained on a smaller robot to goal distance and transferred to a much larger distance in a zero-shot manner. Here, the robot always starts from the location (5, 5) (marked as R in the figure), and the goal is kept in the remaining 120 cells one by one. In both images, a green cell represents that the robot was able to reach the goal placed at that cell and a red cell represents the robot could not reach the goal.

However, the overall trend on all domains remains the same: ORLA is able to do some meaningful zero-shot transfer in both CB and MDB, but CNN-based network fails to do so.

### 4.2   Case Study 2: Robot Navigation

In this second case study, we experiment with two domains set in the context of robot Navigation in a grid world. In the first domain (Navigation), the QoI that we varied is the distance between the robot and the goal location. In the second domain (Pizza), a robot has to reach a goal with a choice to collect some Pizzas (sub-goals) to get a better reward. Here, the QoI, which we vary from train to test environments, is the number of sub-goals (Pizzas) available (size-transfer task). The lack of any natural dynamics and inclusion of sparse rewards separates this case study from the earlier one (which has gravity as natural dynamics).

#### 4.2.1   Domain 3: Navigation (Nav)

The Navigation (Nav) domain has two objects: a robot and a goal. The aim of the robot is to reach the goal in a 2D $11 \times 11$ grid. The Agent gets a 0 reward when at goal (and the episode ends) and -1 otherwise. At each time step, the (train and test) environment returns an image that represents the grid (without any grid lines). During both training and testing, the robot always starts from the center location (5,5).

Using this domain, we want to study whether a robot can learn to reach a goal kept at a distance much farther than what is seen during training. For this, during training, the location of the goal is kept closer to the robot by randomly sampling the goal locations from the coordinate range $(5 \pm 2, 5 \pm 2)$, i.e., the maximum distance between the goal from the robot is 4. At test time, we again keep the robot at the center of the grid $(5, 5)$ and sample the goal anywhere in the $11 \times 11$ grid, i.e., with a maximum distance of 10.

**Results:** Figure 3(d) shows the zero-shot transfer results on this domain. We see that ORLA performs equally well as compared to the G-GAT, while CNN performs much worse than random. We suspect CNN's performance to be due to overfitting, since it is otherwise able to train well on the training environments.

To further analyze the performance, we additionally check the goal coverage of the robot by placing the robot at the center of the grid (5,5) and placing the goal at all the other 120 locations one by one. Figure 4 shows the goal coverage results for ORLA and CNN policy. Here, each green cell represents that the robot was able to reach a goal kept at that cell when starting from location (5,5). We can see that the goal-coverage

of ORLA is 100% whereas CNN could only reach those cells that were seen during the training (with an exception of a couple of additional grid cells).

### 4.2.2   Domain 4: Pizza

We create the Pizza domain to test our model's capabilities on the size-transfer task, where variation in size represents the number of sub-goals available to the robot. The domain is a $6 \times 6$ 2D grid world (represented as an image without any grid lines) with three types of objects: Robot, Goal, and Pizza, placed at different locations. The goal of the robot is to reach the goal location while collecting as many pizzas as possible. Reaching the goal gives a $+5$ reward (ending the episode), collecting a Pizza gives a reward of $+25$, after which the pizza changes color to red and can not be collected again, and -1 otherwise. Note that the robot can directly reach the goal without collecting any Pizza, but for the best reward, it will have to solve a travelling salesman problem with a fixed final node (goal). Thus, this is quite a challenging domain for the network. We create two training environments, one with a single Pizza and another one with two Pizzas. We tested on three environments with 3, 4, and 5 Pizzas, respectively. The location of the robot, all Pizzas, and the goal are sampled randomly at the start of each episode for both training and testing.

**Results:** Figure 3(c) shows the results of zero-shot transfer on the three test environments. We again notice that CNN fails to transfer. ORLA performs worse than random in the initial part of the training and trains slower than G-GAT but finally gets close to it. We hypothesize that the slow learning behavior of ORLA is possibly due to the noise in the output of the object extractor and/or in the k-Means clustering used for assigning the type to each object.

In summary, in case study 2, we study long-horizon sparse-reward environments. We find that these are more challenging to both ORLA and CNN, but ORLA consistently outperforms CNN, showcasing its better transfer capabilities. Our results also indicate that the lack of good performance may sometimes be due to incorrect identification of object masks (or their clusters), suggesting a research direction for the future.

**Results on PPO:** Figures 6(c-d) in supplement show results for PPO. In Nav, all methods struggle to do zero-shot transfer including G-GAT. While CNN performs better than RND in training, it never goes above RND in the test environment, whereas ORLA does perform better than RND in the test environment. In the Pizza domain, the trend seen in dueling DQN remains consistent: for zero-shot transfer, G-GAT performs better than ORLA which performs better than CNN. We will release the code and resources on acceptance.

## 5   Conclusion & Future Work

In this work, we tackled the problem of zero-shot policy transfer across domains with variations in underlying quantities of interest. Our approach is object-centric, consisting of three major components: (1) a novel object extractor trained in a self-supervised manner, (2) a symbolic state consisting of object positions and velocities constructed from extracted objects, and (3) a GAT-based module that takes the symbolic state and decodes it into a policy. Our approach effectively generalizes when transferring across the number of objects, motion dynamics, and distance from the goal state, which is demonstrated through an extensive set of experiments in four different domains and a variety of settings. In contrast, a vanilla CNN based baseline fails to do any meaningful transfer in most cases.

Future work includes evaluating ORLA's efficacy in a few shot transfer settings, testing on domains with varying object sizes and shapes, having complicated motion dynamics, and extending our object-centric approach to a model-based RL setting.

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

# A Appendix: Object-Centric Learning of Neural Policies for Zero-shot Transfer over Domains with Varying Quantities of Interest

## A.1 Architectures

**Expert Object-Extractor**: It takes as input an image with 3 channels (RGB) and outputs a set of K Gaussian masks with their means at the objects' centers. We use the Attention U-Net architecture as proposed in the original work Oktay et al. (2018) for our encoder. It takes a (50, 50, 3) size preprocessed image as input and downsamples it using four convolution blocks with 32, 64, 128, and 256 hidden channels, with 2x2 max-pooling between consecutive layers. It then upsamples the output using convolution blocks with 128, 64, and 32 hidden channels. The last convolution block has output channels equal to the number of objects. We use the learning rate of 5e-4. The output of the Attention U-Net is passed through the Gaussian module to get a Gaussian mask for each channel in the extractor output. It creates a Gaussian mask with its mean at the object's center and fixed $\sigma_x$ and $\sigma_y$ of 2.5. Hadamard Product of the Gaussian masks with the original image gives objects' content, which are used to regenerate the original image by combining them at their respective positions on the background. We use a seed value of 4 for both Numpy and PyTorch.

**YOLOv5**: We use the YOLOv5 model as proposed in the work by Jocher et al. (2022) with their default hyperparameters. The outputs of the Expert Object-Extractors are used to create a labelled dataset to give supervision to fine-tune the pretrained YOLO model.

**Network Architecture**: Our network uses a GAT based architecture to encode the *symbolic-state-graph* in a fixed-dimension (128 dim) feature vector. The symbolic-state-graph is passed twice through graph attention networks. Max pooling over the node features gives the fixed-dimension global embedding vector, which is used to train the downstream RL network.

**Dueling DQN network**: The 128-dimension global embedding is passed through two MLPs to learn the advantage and value functions. The Advantage network consists of two layers with an output dimension equal to the size of the action space. The Value network consists of 2 layers with output of unit dimension. Both the MLP networks have 128 units in the hidden layers and use LeakyRelu non-linearity. For the training, we used the epsilon-greedy method with an initial exploration rate equal to 1, which decreases by a factor of 0.01 after every episode. We train using the Adam optimizer with a learning rate of 0.00025.

**PPO network**: The 128-dimension global embedding is given as input to both the Actor and Critic networks of PPO. The Actor part consists of two MLP layers with 64 units each (with tanh non-linearity), followed by the final layer with an output size equal to the Action Space of the environment. The value network part also has two MLP layers with 64 units each (with tanh non-linearity), followed by the final layer with a single output that gives the value of the state as output.

## A.2 Training

We train all models on a cluster of Quadro P5000 GPUs with 16GB GPU memory and 128 GB RAM. We train each of our Expert Object-Extractors with a maximum time limit of 4 hours and take the checkpoint with the least MSE on a validation dataset. These checkpoints are then used to generate labels for the training dataset to provide supervision to YOLO. We fine-tune YOLO for a maximum of 90 epochs with a maximum time limit of 1 hour time. It takes around 5 minutes to learn clusters in the K-Means clustering module. We give a maximum of 24 hours to train the policy network or other baselines for a satisfactory training time.

Table 1 gives the number of frames used for training the Expert Object-Extractor for different environments.

## A.3 Detailed Dueling DQN Results

Figure 5 shows both the training and test results on various domains when using dueling DQN as the base RL algorithm. See Experiments and Results section in the main paper for explanation on these results.

| Environment | Instance | #Episodes (Train) | #Frames (Train) | #Episodes (Val) | #Frames (Val) |
|---|---|---|---|---|---|
| Column Ball (CB) | 1 | 25 | 23443 | 10 | 9438 |
| | 2 | 25 | 12996 | 10 | 5214 |
| | 3 | 25 | 8961 | 10 | 3557 |
| Multitype Diagonal Ball (MDB) | Uses CB's extractor | | | | |
| Navigation (Nav) | - | 25 | 6770 | 10 | 3495 |
| Pizza (Pizza) | 1 | 250 | 12972 | 100 | 5156 |
| | 2 | 250 | 13482 | 100 | 5262 |

Table 1: Table shows the number of episodes and frames used to train experts in various domains. For MDB, we do not train any experts. Rather, we use the experts trained on Column Ball

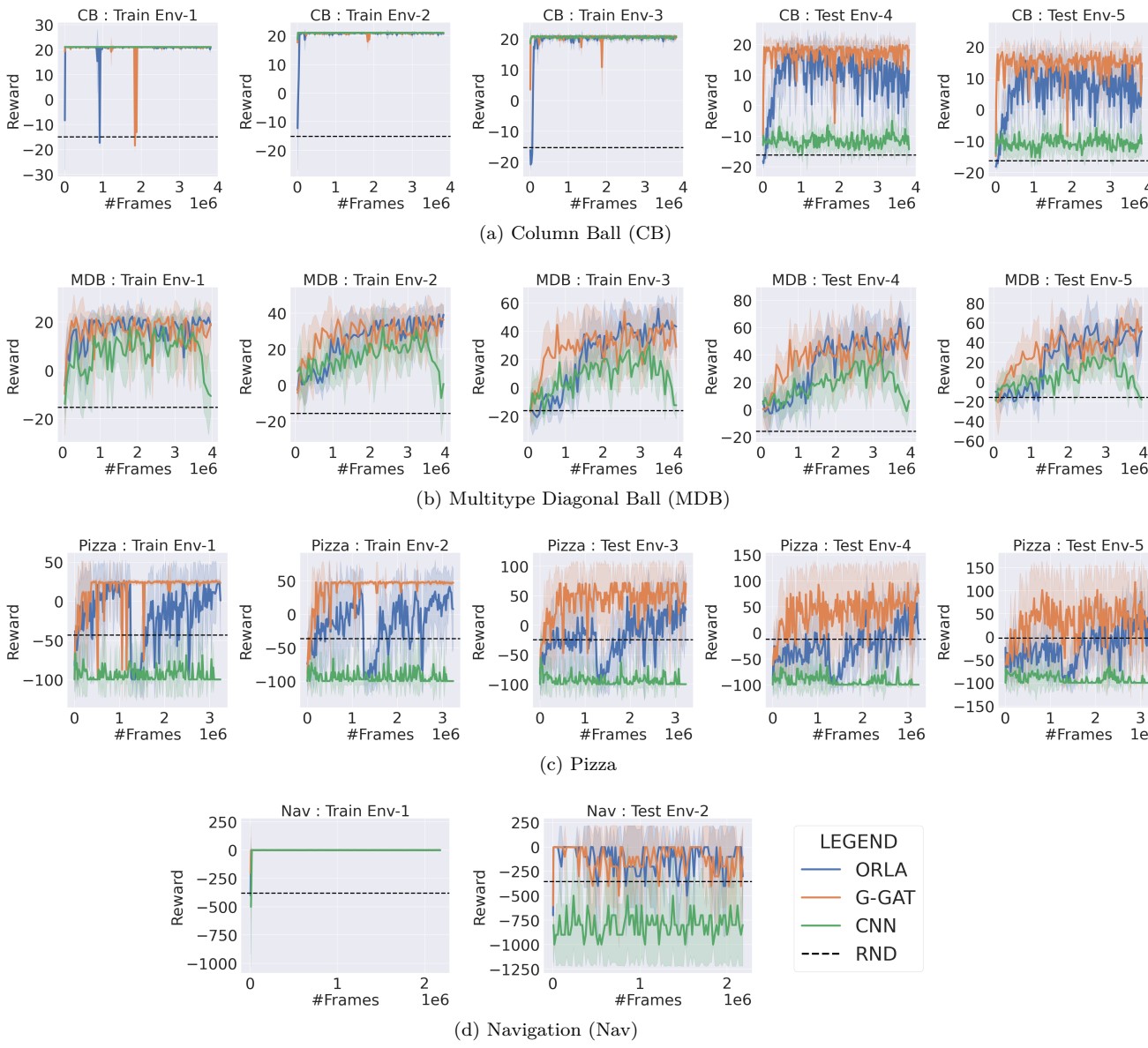

(a) Column Ball (CB)

(b) Multitype Diagonal Ball (MDB)

(c) Pizza

(d) Navigation (Nav)

Figure 5: Figure showing performance (average reward vs #frames) of various methods on all domains on each of the train and test environments for dueling DQN.

## A.4    Detailed PPO Results

Figure 6 shows both the training and test results on all domains when PPO is used as base RL algorithm. The dashed line represents the reward of a uniform random policy over 500 episodes to counter stochasticity in the policy. For all other methods, we take an average of 10 episodes. The first thing we notice is that, in general, training across all methods is unstable as compared to their Dueling DQN counterparts. We attribute this to the inherent nature of PPO. However, the overall trends among results remain the same as in Dueling DQN. We explain these next,

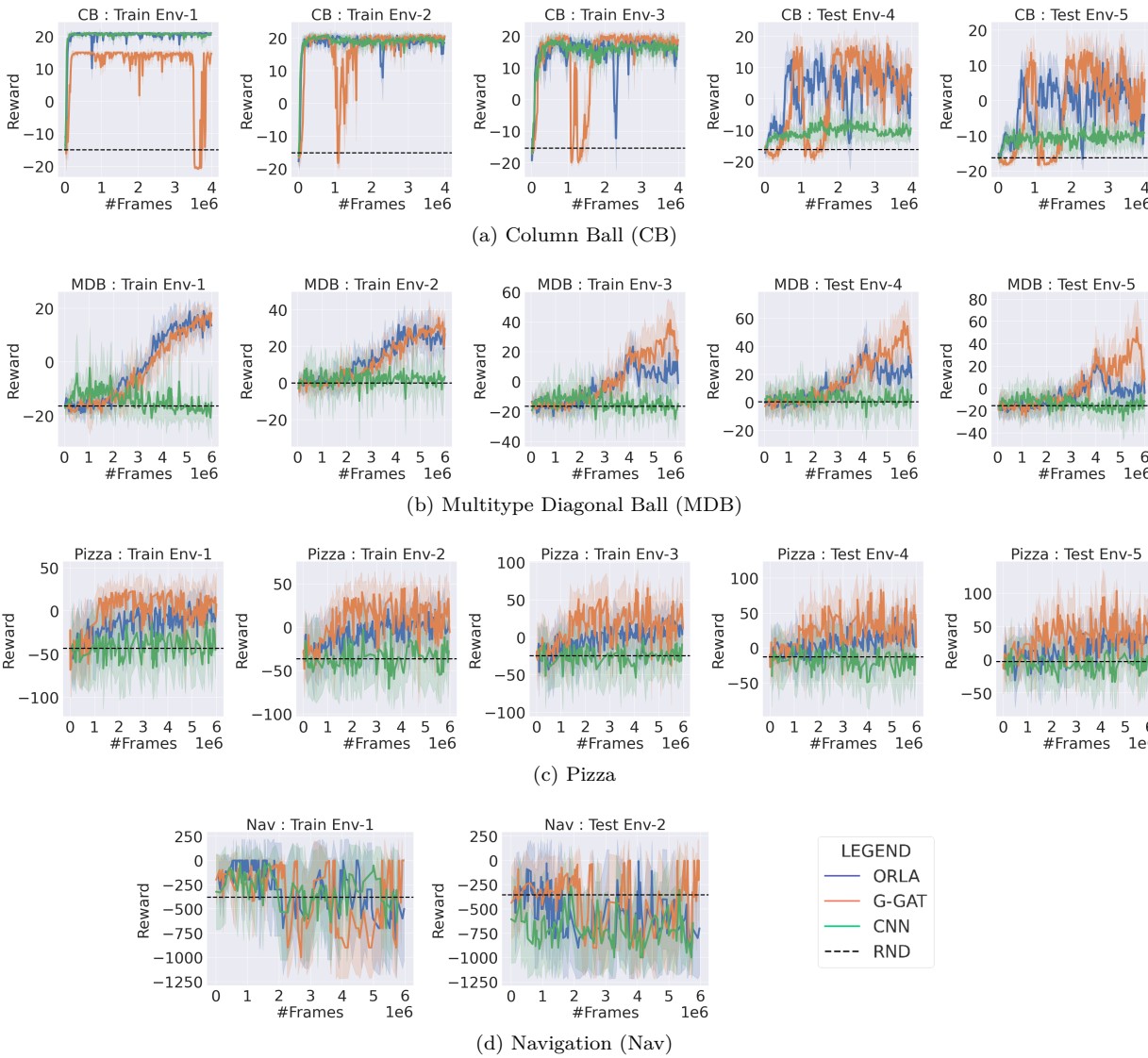

Figure 6: Figure showing performance (average reward vs #frames) of various methods on all domains on each of the train and test environments for PPO

**Column Ball (CB):** For Column Ball (CB), all methods eventually train to get the best possible reward; however, similar to Dueling DQN, CNN based network is not able to do zero-shot transfer on the test environments, whereas ORLA is able to perform much better than the CNN based network, almost at par to G-GAT.

**Multitype Diagonal Ball (MDB):** For Multitype Diagonal Ball (MDB), CNN fails to train, whereas ORLA is able to perform much better than the random policy; however, it is much below the G-GAT. Since

we used the same object-extractor as that in Dueling DQN counterpart for MDB, where ORLA performed at par with G-GAT, we believe the performance drop is because of the use of PPO. While hyperparameter tuning should improve the results, we did not do it in the spirit of developing domain independent algorithm.

**Navigation (Nav):** For Navigation, we observe that for the initial part of the training, all methods train well. However, eventually, for all methods, the training reward starts to fluctuate as the training progresses. During testing CNN-based method performs worse than the random most of the time. This is similar to the behaviour in Dueling DQN, but there are much higher fluctuations in PPO as compared to Dueling DQN. While ORLA also shows fluctuations, it, in general, performs a little better than the CNN-based network. G-GAT performs the best among all but also shows fluctuations during training, sometimes even below random.

**Pizza:** It is the toughest of all domains. While Dueling DQN also showed fluctuations in performance as we continued training, the fluctuations are much more severe for PPO. In general, all methods start training slowly. The overall trend, as seen in Dueling DQN, remains the same, i.e., G-GAT performs better than ORLA, which in turn performs better than CNN. The CNN, in this case, performs as good as random.

### A.5    Results on using existing methods for expert object extractor

We started with using existing state-of-the-art methods of unsupervised object extraction as our expert-OE. We tried two current methods: Slot Attention Locatello et al. (2020) and SPACE Lin et al. (2020).

**Slot Attention as expert-OE:** Figure 7 shows the results of using slot attention for unsupervised object segmentation on the Pizza domain in the environment with 2 Pizzas. Each input image is of size 64x64. We use a total of 5 slots: one for each object and 1 for the background. We used a batch size of 64 and trained for 200 epochs on two A100 GPUs in a distributed training setting. We used the publicly available code of slot-attention[4]. We notice that while slot attention is able to reconstruct the image, the objects appear across all slots at the same time rather than each slot having a single object.

**SPACE as expert-OE:** Figure 8 shows the results of using SPACE Lin et al. (2020) for unsupervised object segmentation on the Pizza domain in the environment with 2 Pizzas. We use a grid size (G) of 4, one for each object, and the background component (K) as 1. Similar to slot attention, we used a batch size of 64 and trained for 200 epochs on the publicly available code of SPACE[5]. We notice that while SPACE is able to reconstruct the image fairly well, it fails to assign a bounding box to any of the objects in the scene. Rather, it detects the whole scene as the background.

Since both slot attention and SPACE failed to detect objects in the scene in an unsupervised fashion on our domains, we developed our own object extractor pipeline using expert-OE. Investigating this behaviour of slot attention and SPACE is left for future work.

---

[4]https://github.com/pairlab/SlotFormer
[5]https://github.com/zhixuan-lin/SPACE

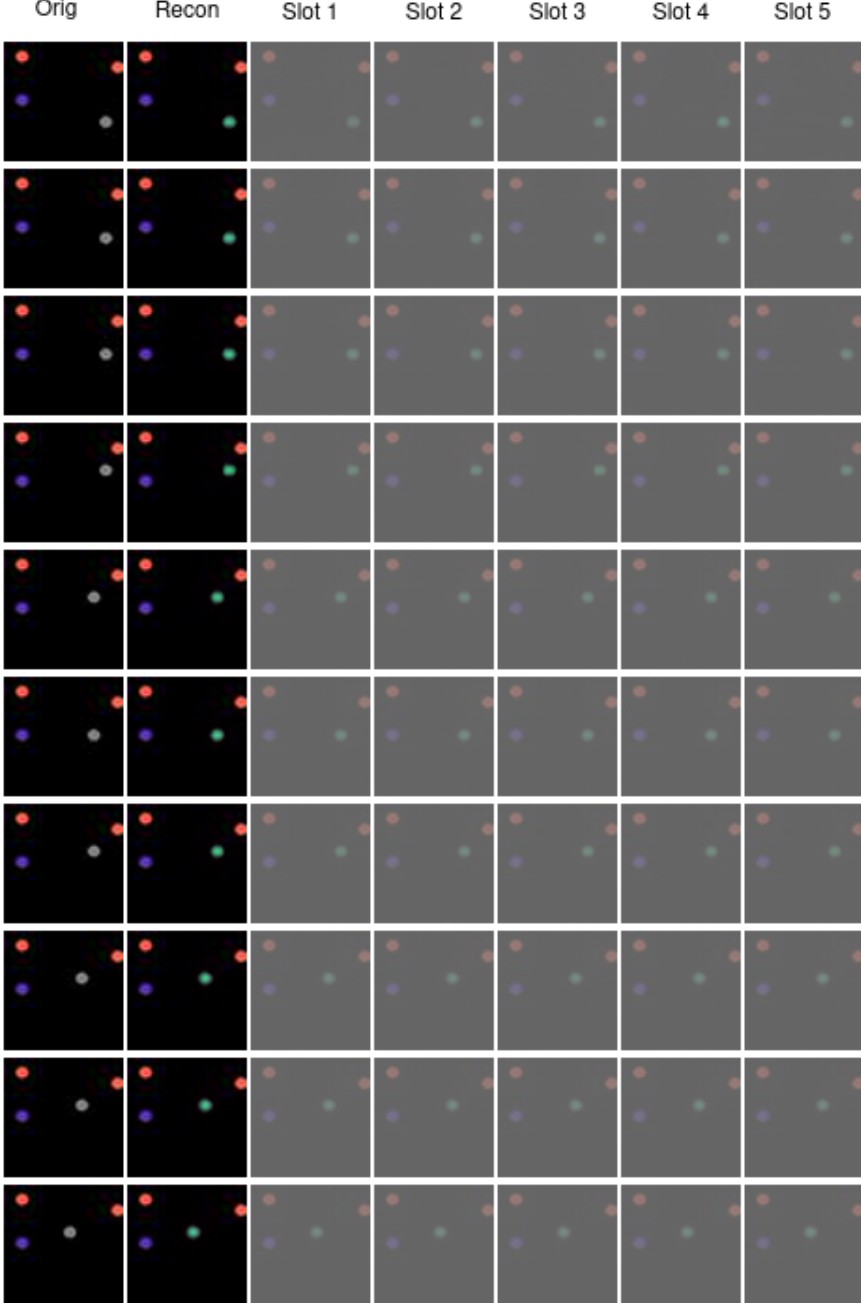

Figure 7: Figure shows the performance of slot attention (Locatello et al., 2020) on Pizza domain for unsupervised object segmentation. The first column is the original image to be decomposed into objects; the second column is the reconstructed image as output by slot attention. The last five columns show the five slots (4 objects and one background). We notice that while slot attention reconstructs the image fairly well, it fails to assign an object to each slot.

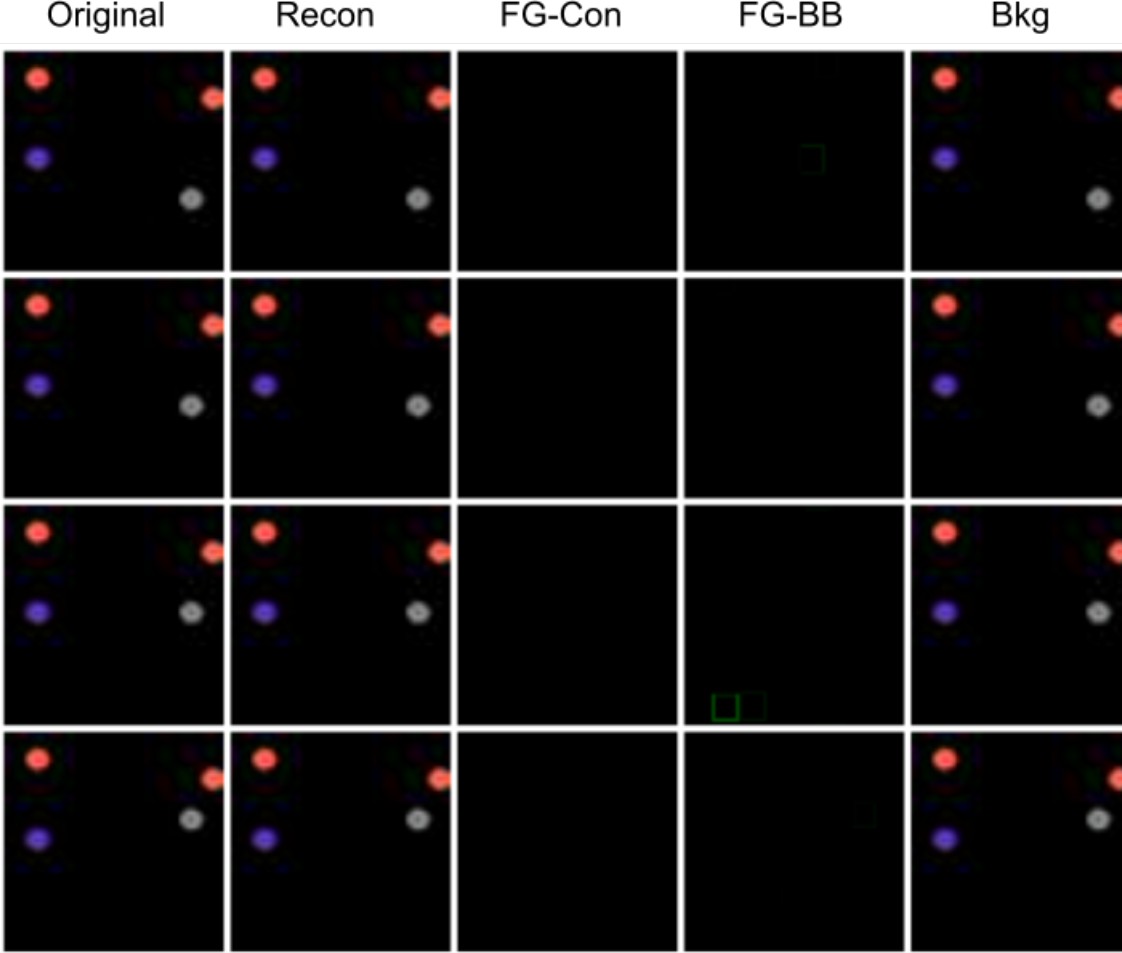

Figure 8: Figure shows the performance of SPACE (Lin et al., 2020) on the Pizza domain for unsupervised object segmentation. The first column is the original image to be decomposed into objects; the second column is the reconstructed image as output by slot attention. The third column is the foreground content (only that part of the scene that contains various objects). The fourth column shows the bounding boxes over various detected objects. The last column shows the background. We notice that while SPACE is able to reconstruct the image fairly well, it fails to assign a bounding box to any of the objects in the scene. Rather, it detects the whole scene as the background.

