# OpenReview forum: "Object-Centric Learning of Neural Policies for Zero-shot Transfer over Domains with Varying Quantities of Interest"
_TMLR — Rejected by TMLR_

### Review · Reviewer_LZZR · 2024-06-10

**Summary Of Contributions:**

This work presents an object-centric visual reinforcement learning method for training agents that can potentially zero-shot generalize to variations in quantities of interest in multi-object environments such as the number of objects or their attributes. The method, named Object-centric Reinforcement Learning Agent (ORLA) consists of the following main stages:

(1) Learning an object-centric representation of images from the set of training environments. This is done in a two-step process where first, a set of structured auto-encoders are trained on images from the corresponding set of training environments separately. These produce a joint dataset that is used to finetune a single pre-trained YOLO model.

(2) Training an online RL policy on the YOLO-extracted object-centric image representations. The extracted objects form a fully-connected graph which is then fed to an attention-based GNN representing the policy/value.

The authors perform an empirical analysis on a set of 2D-grid multi-object game and navigation environments. They test their method’s performance and generalization capabilities compared to a baseline learning directly from raw images using a CNN-based network for the policy/value. The results show superior zero-shot generalization capabilities of ORLA policies compared to the CNN baseline for various quantities of interest.

**Audience:**

Yes

**Broader Impact Concerns:**

No issues here.

**Claims And Evidence:**

No

**Requested Changes:**

**Related Work**

I am not sure I understand your choice of focus in the related work. Object-centric (model-free) reinforcement learning from images would be the most relevant in my opinion, and more specifically work that studies generalization in multi-object environments. This is currently a relatively small field of study yet I believe you are missing many of the most relevant papers in your overview.

*Object-centric Model-free RL from Images*
- [SMORL](https://openreview.net/forum?id=xppLmXCbOw1), “Self-supervised Visual Reinforcement Learning with Object-centric Representations”, Zadaianchuk et al. (2021)
- [OCRL](https://arxiv.org/abs/2302.04419), “An Investigation into Pre-Training Object-Centric Representations for Reinforcement Learning”, Yoon et al. (2023). Note: Studies generalization to various quantities of interest.
- [ECRL](https://openreview.net/forum?id=uDxeSZ1wdI), “Entity-Centric Reinforcement Learning for Object Manipulation from Pixels”, Haramati et al. (2024). Note: Studies generalization to number of objects, includes compositional generalization theory.

*Object-centric Model-based RL from Images*
- [HOWM](https://arxiv.org/abs/2204.13661), “Toward Compositional Generalization in Object-Oriented World Modeling”, Zhao et al. (2022). Note: Studies generalization to various quantities of interest, includes compositional generalization theory.
- [FOCUS](https://arxiv.org/abs/2307.02427), “FOCUS: Object-Centric World Models for Robotics Manipulation”, Ferraro et al. (2023)
- [NCS](https://arxiv.org/abs/2303.11373), “Neural Constraint Satisfaction: Hierarchical Abstraction for Combinatorial Generalization in Object Rearrangement”, Chang et al. (2023). Note: Studies generalization to number of objects.

*Object-centric RL from States*
- [“Policy Architectures for Compositional Generalization in Control”](https://arxiv.org/abs/2203.05960), Zhou et al. (2022). Note: Studies generalization to number of objects and compositions of tasks, includes compositional generalization theory.
- [“Compositional Multi-Object Reinforcement Learning with Linear Relation Networks”](https://arxiv.org/abs/2201.13388), Mambelli et al. (2022)

*Generalization of Object-centric Representations*
- [“Provable Compositional Generalization for Object-Centric Learning”](https://openreview.net/forum?id=7VPTUWkiDQ), Wiedemer et al. (2024). Note: Studies generalization to various quantities of interest, focus on compositional generalization theory.

**Baselines**

*Object-centric Baseline Method*

A comparison to object-centric RL methods is missing. The CNN baseline is important but does not serve as a comparison with state of the art methods that incorporate inductive bias similar to the proposed method. Why did the authors choose [Zambaldi et al. (2019)](https://openreview.net/forum?id=HkxaFoC9KQ) as an attempt for a baseline? There is much more recent and relevant previous work with publicly available code, e.g. [https://github.com/martius-lab/SMORL](https://github.com/martius-lab/SMORL), [https://github.com/jsikyoon/OCRL](https://github.com/jsikyoon/OCRL), [https://github.com/DanHrmti/ECRL](https://github.com/DanHrmti/ECRL). ECRL seems to be SOTA in terms of environment/task complexity and generalization to a larger number of objects. OCRL deals with domains very similar to the ones presented in your work. Since the OCRL paper studies different methods and components, I believe it has many relevant baselines as well.

I request that you compare with at least one of the two: (1) ECRL (2) The best performing method in the OCRL study. (**Critical**)

*Object-centric Representation Method*

As I see it, the main novelty/contribution of this work in terms of the method is the pre-training object-centric representation scheme. Therefore, I believe a comparison and ablation is missing in order to truly assess the value of the proposed scheme. I would request the following:

(1) A baseline model replacement for the expert-OE. You mention that slot-based methods failed to bind one object per slot - this is a common problem with slot-based models. I would suggest trying the [Deep Latent Particles](https://arxiv.org/abs/2205.15821) model used in ECRL which provides explicit positions as well as bounding boxes and seems to work well for downstream multi-object RL tasks. Alternatively you can try the [SCALOR](https://arxiv.org/abs/1910.02384) model used in SMORL that operates on videos and includes object tracking which your method could potentially make use of. (**Critical**)

(2)  An ablation replacing the fine-tuned YOLO component with a single unsupervised expert-OE model trained on all training environments. This would show that using multiple models is indeed necessary compared to a simpler training of a single unsupervised model on multiple tasks. (**Critical**)

*Non Object-centric Baseline Method*

While CNN provides an unstructured baseline, I believe it does not properly isolate the contribution of object-centric structure to solving the tasks as it learns perception concurrently with control which may or may not be harder than the setup in ORLA where these are learned in separate stages. I believe using a pre-trained unstructured representation (e.g. a standard autoencoder) as a baseline could shed more light on this point.

*Object Extraction Visualization*

You have figures showing how slot-attention and SPACE fails at object segmentation but do not present figures visualizing the proposed expert-OE segmentation results. Such a figure would help to better understand the performance of your method.

**Experiments**

*Experiment Description*

The experiments’ descriptions lack details that are important for understanding the results: (1) Environment images of pizza should be in the main text. (2) The natural dynamics in the “Balls and Pedal” environments are not clear; Do the balls interact with each other in the DB environment? What happens when they hit a wall or the ground? Does the speed remain constant? (3) Environment videos of the agent solving the tasks would really help understand both the environments and the generalization behavior of the agent. (**Critical**)

*Column-Balls Experiment*

The CB experiments you chose do not isolate the aspects of generalization required to succeed in the given tasks in my opinion. The QoIs are: (1) The number of objects (2) Object’s x position (column) (3) Object’s velocity. Can the CNN generalize to new positions with the same number of objects as in training? Can the CNN, given training data that includes all columns, generalize to an increasing number of objects? How does the CNN suddenly generalize to 4 and 5 objects in the MDB environment when it failed in the CB experiments? Maybe this has to do with the above questions. Could the authors please run experiments that would answer these questions? (**Critical**)

*Other Questions/Remarks*

- The reward curves are very noisy, indicating you might need more than 10 episodes for proper evaluation.
- Why are there no G-GAT results in Figure 3(e)?
- How many random seeds are used for the confidence intervals in the graphs?
- In my opinion, PPO experiments and discussion don’t give much additional insight.

**Environments**

I have a few concerns regarding the complexity of the environments and tasks:

- *Visual complexity*: Since the paper proposes a method for solving tasks in visual domains, it is hard to convince of its relevance to real-world tasks with experiments on such visually simple environments. The environments are 2D and have no noise or occlusion, little to no variation in object color, shape and size, single-color background etc. Evaluating your method on simulated robotic multi-object manipulation environments such as the ones used in SMORL and ECRL and/or the [Object-centric Atari Environments](https://github.com/k4ntz/OC_Atari) (if you wish to remain in the 2D discrete-action setting) would significantly strengthen your claims.

- *Task complexity*: The tasks presented in this work are relatively simple compared to recent work on object-centric RL. The action and state spaces are small and discrete and interaction between objects is limited to agent-object interaction. For instance, the fact that a CNN-based agent is able to somewhat solve the multi-object MDB tasks and even generalize to a larger number of objects suggests that the task is not very challenging.

Evaluating your method on simulated robotic multi-object manipulation environments such as the ones used in SMORL and ECRL and/or the [Object-centric Atari Environments](https://github.com/k4ntz/OC_Atari) (if you wish to remain in the 2D discrete-action setting) would significantly strengthen your claims.

**Method**

Regarding the YOLO finetuning: can the authors please clarify exactly what labels they need in order to finetune the YOLO model and how they acquire them from the expert-OEs?

I have a few concerns about the generality and applicability of the method to more complex environments:

- *History-objects*: your method requires explicit matching between objects in consecutive timesteps, i.e. tracking. This matching is done by choosing the closest object which belongs to the same category. In more visually complex and real-world domains, occlusion or high-density of objects is something that can really hinder the performance of your method as it strongly depends on accurate tracking.

- *Object velocity attribute*: your method assumes that object velocity is the only important temporal attribute to solve the task, which to my understanding, is not always the case. For example in the pizza domain, the velocity is not required at all but maybe the changing of color of the pizza is important. Wouldn’t some type of frame-stacking and appropriate changes to the graph/architecture in this case be a more general way to handle partial observability?

- *Object types*: An assumption you make in your method is that object masks correspond to object types of interest for control, and that they are consistent across time. What if the agent and the objects are assigned to the same cluster? What if an object is not symmetrical in shape and can change orientation so that it appears different in consecutive timesteps? Wouldn’t the type of the object based on clustering of the masks change with time?

I suggest discussing these points in the paper, possibly in a separate “Limitations” section.

**Strengths And Weaknesses:**

Strengths:
- Interesting and possibly novel self-supervised object-centric representation-learning scheme
- Studying transfer of RL policies to changes in a variety of quantities of interest
- Intend to make code and environments publicly available

Weaknesses:
- Missing many relevant and recent papers in the related work
- Lacking relevant baselines although ones exist in recent work
- Experiment design makes it hard to make precise conclusions about generalization capabilities
- Visually simple environments (2D, simple shapes/colors/background)
- Relatively simple tasks (2D, small and discrete state space, small and discrete action space)
- Strong simplifying/environment-specific assumptions

---

### Review · Reviewer_orAN · 2024-06-17

**Summary Of Contributions:**

This paper presents a method for introducing object-centric concepts into reinforcement learning agents. The proposed method first learns an object detection model in a self-supervised way, and then it constructs a so-called symbolic state graph that contains object positions and velocities, and uses a graph attention network (GAT) to reason about actions. The contributions mainly focus on how to allow an RL agent to generalize its policy in a zero-shot manner to tasks with different QoI (Quantities of Interest).

**Audience:**

Yes

**Broader Impact Concerns:**

I do not see any impact concerns.

**Claims And Evidence:**

Yes

**Requested Changes:**

* Discuss more literature on the connection between learning these self-supervised object models and pre-trained object detection models.
* Discuss in more detail / concretely the contribution of (2).
* Change "symbolic" to other terms, unless the authors can provide a very strong argument for using "symbolic state" to refer to positions and velocities of objects.
* Add more domains to support each argument. I would like to see two domains for each argument the authors trying to convey.
* It would be nice, especially for scattered experiment study like this, to have a list of experiment questions at the top. In this way, readers can better follow  the contents without getting lost.

**Strengths And Weaknesses:**

**Strengths:**
* The design of experiments is quite thorough.
* The discussion of case studies is thorough.
* The authors focus on the "extrapolation" ability of the policy.

**Weaknesses:**
* Can authors formulate, define, or describe why we would expect the learned behavior to generalize to a number of objects that the agent has never seen? I think this is a very important concept that requires a very clear statement / description if the whole concept of doing zero-shot transfer makes sense.
* While the explanation of each experiment is thorough, each research question is validated using one domain. This brings into question whether the conclusion can be applied to other domains.
* While I understand that the authors follow the paradigm of discovering objects in an unsupervised manner, I would like to ask authors to think about this question and also at least address this to some extent in the writing: Why do you need to do unsupervised discovery of objects? So far in the literature, I barely see any tested domains in this line of work that require domain-specific training of objects — most of the pixel domains seem to be able to infer objects both based on colors and their movements, while slightly more complicated domains are also common object concepts that pre-trained vision models can handle (and their generalization is very good, and they can handle a large variety of objects). Are the domains really sufficient to show that this technique makes sense? Or is it the case that most domains that this proposed methodology can handle are also solvable by just running pre-trained object detection models or segmentation models? Are there any domains that really cannot be handled with existing pre-trained detection model but can shine with your way of discovering objects?
* Regarding the last second contribution, it is unclear and not explicitly stated how this formulation is different from other prior works, such as [1].
* I would argue that the use of the term "symbolic" is ambiguous — "symbolic" usually refers to either variables in programs or semantics. I would call it "geometric" or "cartesian", as the encoded information, positions and velocities, are all physical information.

[1] Graph Inverse Reinforcement Learning from Diverse Videos. Sateesh Kumar, Jonathan Zamora, Nicklas Hansen, Rishabh Jangir, Xiaolong Wang

---

### Review · Reviewer_ibWc · 2024-06-29

**Summary Of Contributions:**

In this work, the authors propose a novel method to allow better RL generalization over variations in quantities of interest (e.g. number of balls in the environment). The proposed method, Object Centric Reinforcement Learning Agent (ORLA) has 3 phases: 1. learn object extractor that can produce object masks. 2. A YOLO model is trained on these object masks to produce bounding boxes. 3. the output of the YOLO model is put into a GAT to produce generalizable representation that is feed into a policy network. The authors provide empirical results that show such a setting can lead to better generalization compared to baselinse that directly use perceptual input.

**Audience:**

Yes

**Claims And Evidence:**

No

**Requested Changes:**

The paper is very interesting, but can benefit from more comprehensive experiments, such as comparing to stronger baselines. Additionally, it can be good to have more ablations experiments to understand whether the proposed method is sensitive to hyperparameter choices or modifications of the proposed technical components. The authors provided training time for the proposed method, might be good to add a short discussion to more explicitly compare its training time and the baselines.

**Strengths And Weaknesses:**

Strengths:

**Novelty**:
- the proposed methods (a learned system to convert visual features into more object-centric ones) and presented empirical results can be considered novel contributions.

**Clarity**:
- Overall the paper is clear, with minor issues

**Significance**:
- converting perceptual features into more object-centric ones seem reasonble especially if we have knowledge of the task that these objects are crucial for the optimal behavior of agents. Understanding how changing the feature space will affect the difficulty of transfer/generalization is an important question and adds to significance of results presented in this paper.

Weaknesses:

**Quality**:
- Since a lot of experiments are done in new benchmark environments (I mean, for example, DQN and its variants are mainly done on Atari, and the environments studied in this paper are different). It is interesting that the proposed method can do better on these new environment, however, I am not sure if the baselines are indeed perform really bad or if they are just not set up correctly (e.g. due to suboptimal hyperparameters). While robust algorithms should not rely on excessive hyperparameter tuning, I wonder if the authors have done experiments or sanity checks to show that the baselines are indeed performing at a reasonable level?

**Significance**:

- I am not entirely convinced that the proposed method will have better generalization compared to, say, a baseline that directly learns from visual features. For example, in the paper it is mentioned that the object extractor learns from datasets from the domain with variations. Now, if we can assume access to different environments/data in the domain of interest, then it might be reasonable to compare to stronger baselines that are designed for such a transfer/generalization setting. Are the baselines used in the current paper strong baslines? Or are they a bit naive/performing in an unfair setting?
- It seems to me the proposed method will require expert knowledge in identifying what objects are relevant to the task (correct me if I misunderstood). It seems to me this might make the proposed method harder to use compared to a baseline that operates directly on visual features. Additionally, the proposed method brings additional networks and complexity to training, which can add to the difficulty.

---

### Decision · Action_Editor_vxxn · 2024-09-20

**Recommendation:** Reject

**Comment:**

All reviewers have pointed out significant drawbacks with the manuscript on experimental setups, thoroughness of conceptual and empirical comparisons to prior work, and strong assumptions in the approach. The authors have unfortunately not responded to these claims.

**Audience:**

The target audience would be robot learning and adjacent fields, but the paper as it stands falls short of being a valuable addition to the literature here.

**Claims And Evidence:**

All reviewers have pointed out significant drawbacks with the manuscript on experimental setups, thoroughness of conceptual and empirical comparisons to prior work. As such, the claims are not supported by convincing evidence.